# Open access policies of leading medical journals: a cross-sectional study

Tim S Ellison,[1] Tim Koder,[2] Laura Schmidt,[2] Amy Williams,[1] Christopher C Winchester[2]

[1]PharmaGenesis London, London, London, UK
[2]Oxford PharmaGenesis Ltd, Oxford, UK

**Correspondence to**
Dr Tim S Ellison;
tim.ellison@pharmagenesis.com,
timsellison@gmail.com

## ABSTRACT

**Objectives** Academical and not-for-profit research funders are increasingly requiring that the research they fund must be published open access, with some insisting on publishing with a Creative Commons Attribution (CC BY) licence to allow the broadest possible use. We aimed to clarify the open access variants provided by leading medical journals and record the availability of the CC BY licence for commercially funded research.

**Methods** We identified medical journals with a 2015 impact factor of ≥15.0 on 24 May 2017, then excluded from the analysis journals that only publish review articles. Between 29 June 2017 and 26 July 2017, we collected information about each journal's open access policies from their websites and/or by email contact. We contacted the journals by email again between 6 December 2017 and 2 January 2018 to confirm our findings.

**Results** Thirty-five medical journals publishing original research from 13 publishers were included in the analysis. All 35 journals offered some form of open access allowing articles to be free-to-read, either immediately on publication or after a delay of up to 12 months. Of these journals, 21 (60%) provided immediate open access with a CC BY licence under certain circumstances (eg, to specific research funders). Of these 21, 20 only offered a CC BY licence to authors funded by non-commercial organisations and one offered this option to any funder who required it.

**Conclusions** Most leading medical journals do not offer to authors reporting commercially funded research an open access licence that allows unrestricted sharing and adaptation of the published material. The journals' policies are therefore not aligned with open access declarations and guidelines. Commercial research funders lag behind academical funders in the development of mandatory open access policies, and it is time for them to work with publishers to advance the dissemination of the research they fund.

## Strengths and limitations of this study

► This manuscript includes a cross-sectional analysis of open access policies of medical journals with a high impact factor, including society-owned journals, from multiple publishers.

► The open access policies of all journals analysed were clarified, and confirmation of our findings was received by email from 97% of the contacted journals.

► Open access policies of the journals and publishers analysed are subject to change, so the information presented here may change in the future.

► By selecting journals with a high impact factor, our analysis does not include prestigious journals from specialised therapy areas and regional or non-English language journals, which may have lower impact factors.

► Some of the journals included in our analysis (eg, *Science*, *Nature*) could be considered as interdisciplinary journals rather than exclusively medical journals.

average of 17 years for research evidence to reach 50% adoption in clinical practice, with the longest delays occurring after successful publication of clinical trial results.[2 3] Implementation of research published using the traditional subscription publication model is hindered by copyright restrictions that prohibit reuse of the published content and paywalls that prevent public access.

Open access publishing has the potential to improve innovation and speed up its adoption. Complete access to research literature encourages viewing of more articles than partial access,[4 5] and open access articles appear to be downloaded more often and receive more citations than subscription articles, indicating a greater academical impact.[6–9] There is also evidence suggesting that open access articles have a broader societal impact based on altmetric data that measure the attention publications receive in the news and social media.[9–11] Depending on the restrictiveness of its licensing, open access can facilitate public and commercial

## INTRODUCTION

Hundreds of billions of US dollars are invested in medical research by governments, charities and philanthropical and commercial organisations each year, with the aim of extending and improving human lives.[1] Publication plays an important role in the dissemination of scientific innovation.[2 3] However, translation of medical research into clinical practice is slow; one study has suggested that it takes an

reuse of research results, which is beneficial for collaboration, education and innovation.[9] Access to the full text of research articles also increases transparency, benefitting the public by helping both doctors and patients to find complete and current evidence to inform treatment decisions, and by preventing potentially harmful decisions being made based on the abstracts of paywalled articles.[9 12–14] The publishing model used by a journal (ie, open access or subscription) has no impact on the quality of articles published.[15 16]

'Open access' is a broad term that encompasses a range of definitions, from 'free-to-read' (full text available to read on demand, without charge to the reader) to 'free-to-read and reuse' (with the additional ability to reuse text, tables and figures in different formats). The Budapest Open Access Initiative,[17] the Berlin Declaration,[18] the Bethesda Statement[19] and open access advocates[20] define 'open access' exclusively as published content that can be read free-of-charge immediately at the time of publication with unrestricted reuse rights providing that the original source is attributed. Therefore, these open access advocates and declarations recommend open access publishing under a Creative Commons Attribution (CC BY) licence, which allows sharing and adaptation of published materials for any purposes (both commercial and non-commercial), subject only to attribution of the original source.[17 21 22] Common alternatives to the CC BY licence include CC BY Non-Commercial (CC BY-NC), which restricts commercial reuse; CC BY No Derivatives (CC BY-ND), which restricts adaptation and CC BY-NC-ND, which restricts both (online supplementary file 1).[21 23]

When a journal offers open access, it has wide scope in the choice of policy or policies it will apply, using one of the Creative Commons licences that allow reuse under specific terms, or offering free-to-read access without a licence.[21] The Directory of Open Access Journals (DOAJ) requires journals indexed in the directory to state on their websites clearly and precisely the terms of use and reuse that readers and authors have when they submit an article or use the published content. DOAJ has a strong preference for the use of Creative Commons licences, especially the CC BY licence.[24]

At prominent journals, open access with a Creative Commons licence is typically facilitated by an article processing charge. Following payment by the research author, institution or funder, articles are usually made available on the journal's website at the time of publication in the publisher's typeset format (Version of Record). Open access articles that do not include a Creative Commons licence at the time of publication typically involve an embargo period before the published articles are freely accessible and may allow access only to the accepted manuscript (a version that has not been edited and typeset by the journal), which is made available on the author's institutional website, PubMed Central or Europe PubMed Central without a requirement for payment. It is noteworthy that the accepted version of a manuscript as well as the Version of Record are sometimes required to bear a Creative Commons licence, often including the -NC and/or -ND clause.[25]

There has been an increasing trend towards open access publishing over the last 20 years, and almost 50% of articles were published open access in 2015.[8] However, a study analysing global health research articles published between 2010 and 2014 showed that 69% of these articles were not freely available on the journal's website and 61% of researchers do not self-archive their work even when journal policy allows them to do so free of charge.[26] Many academical and not-for-profit research funders now require the research they fund to be published open access.[9 27–32] Prominently, the Wellcome Trust and the Bill & Melinda Gates Foundation insist on publishing with a CC BY licence to allow the broadest possible use.[27 29] Commercial research funders, which fund approximately half of all medical research,[1 33 34] have been more hesitant to require open access publishing but now commonly pay for open access when the option is available.[30] Commercial research funders are defined here as pharmaceutical companies and other medical industries that fund research for commercial purposes. The proportion of articles authored by large pharmaceutical companies that were published open access doubled between 2009 and 2016.[35] In January 2018, Shire (now part of Takeda) became the first commercial research funder to require all research manuscripts it funds to be published open access.[36 37] One year later, Ipsen committed to making its published scientific research freely accessible to everyone.[38]

We set out to clarify the open access variants provided by leading medical journals for research in general, and commercially funded research in particular, and establish the availability of the CC BY licence for commercially funded research.

## Methods

Using Journal Selector (Sylogent, Newtown, Pennsylvania, USA), we identified medical journals with a 2015 impact factor of at least 15.0 (accurate on 24 May 2017). To focus on journals publishing original medical research, we excluded journals that only publish review articles. We collected information on the open access variants provided by the included journals from their websites and by email contact when information was missing or unclear, making up to three attempts between 29 June 2017 and 26 July 2017.

For each journal, we recorded the following information:
► For immediate open access, whether a CC BY licence or other Creative Commons licence was provided.
► For delayed open access, the length of embargo period for open access.
► For both immediate and delayed open access, which version of the article would be available (published Version of Record or accepted).

For journals that provided a CC BY licence, we additionally collected information on:

**Table 1** Categorisation of journals based on the most open variant of open access offered

| Category | Version of article available | Embargo period* | CC BY licence offered by the journal? |
|---|---|---|---|
| 1 | Published | None | Yes |
| 2 | Published | None | No |
| 3 | Published/accepted | ≤12 months | No |

CC BY, Creative Commons Attribution.
*None = immediate open access; >0 months = delayed open access.

- ► The requirements for obtaining a CC BY licence (eg, dependence on funding source).
- ► Article processing charges.

Between 6 December 2017 and 2 January 2018, we emailed the journals' editorial offices requesting confirmation of our findings. Once open access variants were recorded, we categorised the most open variant provided by each included journal using our own classification, as shown in table 1.

To gather general information on open access licences and charges available from a larger selection of medical journals, we carried out a search on the DOAJ website (https://doaj.org/search) on 21 February 2019.

### Patient and public involvement
Although patients and/or the public were not directly involved in the design and conduct of this study, patients' perspectives were sought during the reporting of our findings and are included in the online supplementary file 1.

### RESULTS
#### Included journals
Fifty-three journals listed in the Journal Selector database had a 2015 impact factor of at least 15.0 (figure 1). After 16 review journals and two non-medical journals were excluded, 35 journals from 13 publishers were included in this analysis. Of the 15 journals that were contacted to clarify information that was missing or unclear, 14 replied with clarification. Once all information was collected and tabulated, we received confirmation of our findings from 34 (97%) of the 35 journals.

#### Open access variants offered
Proportions of journals in each category of the most open variant of open access are shown in figure 2A. Immediate open access with a Creative Commons licence was provided by 21 (60%) of the 35 journals analysed. The types of Creative Commons licence available from these 21 journals under different circumstances were: CC BY from 21 journals (100%), CC BY-NC from four journals (19% of all journals offering CC BY) and CC BY-NC-ND from 18 journals (86% of all journals offering CC BY).

When the 35 analysed journals were categorised by impact factor, immediate open access with a CC BY or other Creative Commons licence was provided by 10

(66%) of the 15 journals with an impact factor between 15.0 and 19.9, and 3 (30%) of the 10 journals with an impact factor over 30.0 (figure 2B).

All 14 journals, from six publishers, that did not provide open access with a Creative Commons licence provided access to different versions of the article either immediately, after a 6 month embargo period or after a 12 month embargo period under different circumstances (table 2).

### The cost of open access with a CC BY licence
Of the 21 journals that offered a CC BY licence, 19 (90%) disclosed article processing charges on their websites. Across these journals, charges ranged from US$3000 to US$5000; the most common article processing charge was US$5000 (in 13 (62%) of journals; figure 3). Of the six journals disclosing an article processing charge of less than US$5000, five had an impact factor of less than 20.0, indicating that the cost of article processing charges may depend on impact factor. Details of the fees charged by the remaining two journals (10%), *Science* and *Science Translational Medicine*, were not available from their

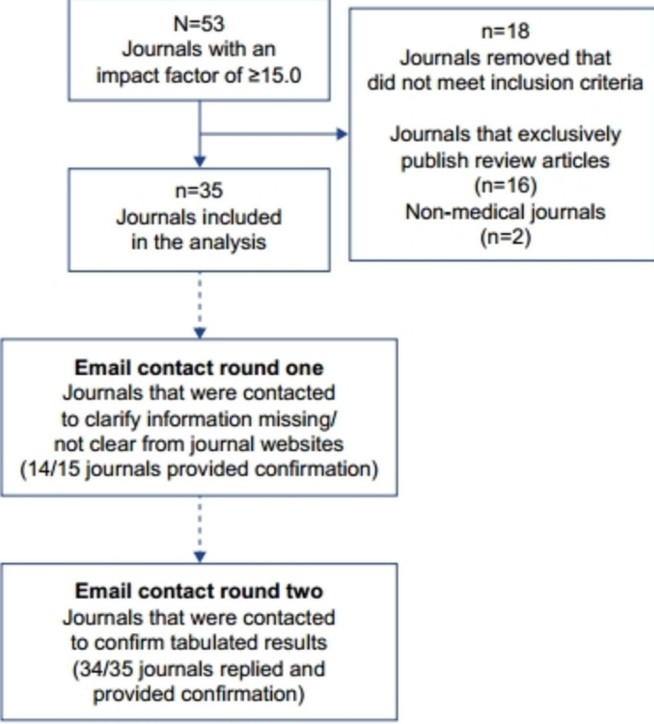

**Figure 1** Flow chart of journals included in this study.

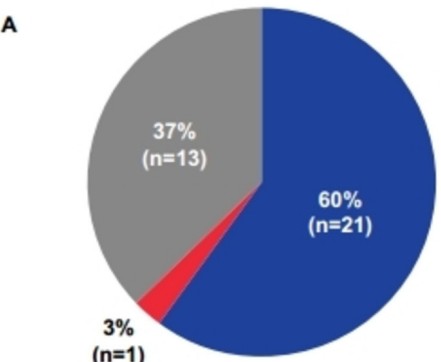

- ■ Category 1: published version of record available upon publication with a CC BY licence
- ■ Category 2: published version of record free to read upon publication (no Creative Commons licence)
- ■ Category 3: published version of record or accepted version first available 6–12 months after publication (no Creative Commons licence)

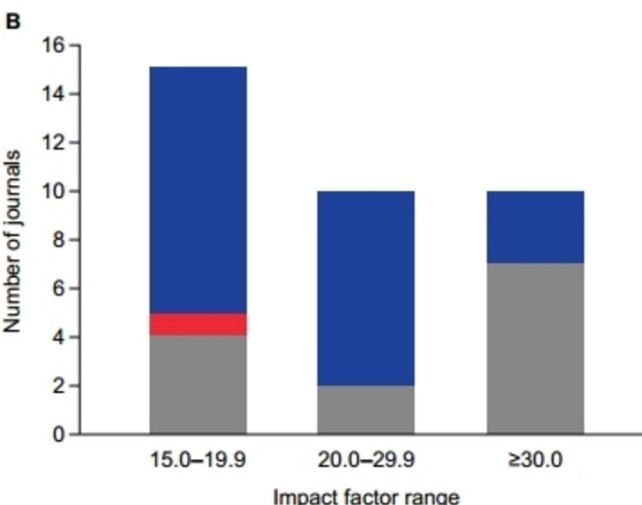

**Figure 2** Medical journals categorised by impact factor and their most open variant of open access available (n=35). (A) Impact factor ≥15.0 and (B) Impact factors 15.0 to 19.9, 20.0 to 29.9 and ≥30.0. CC BY, Creative Commons Attribution.

websites because the details were only provided when the article was accepted.[39]

### Relationship between funding source and the availability of open access variants

Table 3 shows the open access policies of the journals offering open access with a CC BY licence. Of the 21 journals listed, 20 journals allowed open access with a CC BY licence for research funded by specific non-commercial organisations, and only *The BMJ* offered it to any organisation who required it, regardless of the nature of the funding source.

### Availability of open access from a larger selection of medical journals

Of 713 medical journals indexed in the DOAJ database on 21 February 2019: 689 (96.6%) supported publishing

with a Creative Commons licence and 24 (3.4%) supported publishing with the publisher's own licence; 227 (31.8%) journals charged article processing charges for open access and 257 (36.0%) journals listed CC BY as their most restrictive licence regardless of whether there was an associated article processing charge. Of the 257 journals that allowed open access publishing with a CC BY licence, 108 (42.0%) charged an article processing charge for the opportunity and two (0.8%) did not have available information on publication charges.

### DISCUSSION

Here, we present a cross-sectional analysis of open access policies of medical journals with a high impact factor, including society-owned journals, from multiple publishers. We met our objective to clarify the open access policies of all journals analysed and received confirmation of our findings by email from 97% of the contacted journals. We found that all leading medical journals in this study provided some form of open access, but there was little consistency across their policies. Over half of the included journals provided a CC BY licence; however, with the exception of one journal, this option was available only to authors funded by non-commercial organisations. One journal (*The BMJ*) allowed authors to obtain a CC BY licence when the work was supported by funders who required its use. Therefore, if commercial organisations, such as pharmaceutical companies had a policy that required open access with a CC BY licence, *The BMJ* would currently be the only compliant medical journal with an impact factor greater than 15.0. At the time of our analysis, no commercial research funder required open access with a CC BY licence. However, the company at which the analysis was performed, Oxford PharmaGenesis, has since updated its publication policy to require open access with a CC BY licence for the research it funds.[40]

Limitations of this study are that we investigated journals listed in the Journal Selector database with an impact factor of at least 15.0, and that, because impact factors and the open access policies of journals and publishers are subject to change, the information may change in the future. The validity of the impact factor metric is contentious, and its use in this analysis may have led to exclusion of prestigious journals from specialised therapy areas and regional or non-English language journals that have impact factors under 15.0. Furthermore, some of the journals included in our analysis (eg, *Science*, *Nature*) can be considered interdisciplinary journals rather than exclusively medical journals. Although our study covers only a small number of journals, extending such a manual analysis to a greater number of journals without loss of detail and verification of all results would take more time and increase the scope of the study. If more extensive mining of journal metadata becomes feasible, this study could be more easily repeated for a bigger cohort of journals. To gather general information on open access licences and charges available from a larger selection of

**Table 2** Access policies of journals with high impact factors that do not provide open access with Creative Commons licences

| Publisher | Organisation status | Journals included (n=14) | Open access variants available* | |
| --- | --- | --- | --- | --- |
| | | | **Embargo period†** | **Version of article available** |
| American Association for Cancer Research Journals | Non-profit society | *Cancer Discov* | None | VoR‡ |
| | | | 6–12 months | Accepted |
| American College of Physicians | Non profit society | *Ann Intern Med* | 6 months | Accepted |
| American Medical Association | Non-profit society | *JAMA* | None | VoR§ |
| | | | 6 months | VoR |
| Massachusetts Medical Society | Non-profit society | *N Engl J Med* | 6 months | VoR |
| Nature Publishing Group | Commercial | *Nature*; *Nat Biotechnol*; *Nat Cell Biol*; *Nat Genet*; *Nat Immunol*; *Nat Mater*; *Nat Med*; *Nat Methods*; *Nat Neurosci* | 6 months | Accepted |
| Wiley-Blackwell | Commercial | *World Psychiatry* | 12 months | Accepted |

*Available under the terms specified on the journal website.
†None = immediate open access; >0 months = delayed open access.
‡On payment of US$3500 AuthorChoice fee.
§Available to read on JAMA Network Reader.
VoR, version of record.

medical journals, we carried out a search on the DOAJ website on 21 February 2019. Unlike our manual analysis of medical journals with a high impact factor, the search of medical journals indexed in the DOAJ included only those that met the DOAJ criteria to be considered an open access journal. Therefore, it is not surprising that

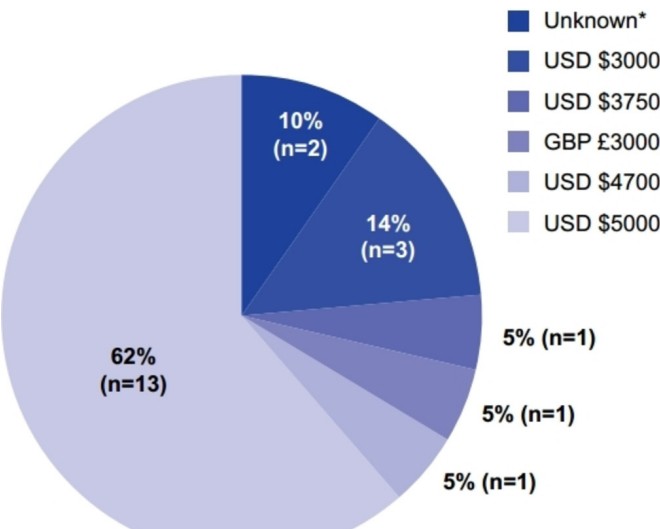

**Figure 3** Article processing charges of journals that offer immediate open access with a CC BY licence (n=21). *Details on processing fees are provided at acceptance.[39] CC BY, Creative Commons Attribution.

the proportion of journals that provided open access with a Creative Commons licence in our manual analysis (60.0%) was lower than that found in the DOAJ search (96.6%). To our surprise, the proportion of journals that provided open access with a CC BY licence was higher for the journals in our manual analysis (60.0%) than for the journals identified in the DOAJ search (36.0%). However, the DOAJ currently only lists one licence for each journal and asks publishers to choose the most restrictive licence, so there is a possibility that the CC BY licence is available from a greater proportion of medical journals indexed in the DOAJ. In our manual analysis, all included medical journals that provided open access with a CC BY licence required payment of an article processing charge, but less than half of the medical journals in the DOAJ charge for a CC BY licence. This finding suggests that medical journals with a high impact factor charge more for publishing open access with a CC BY licence than the average medical open access journal. Information on whether the availability of the CC BY licence is dependent on the funding source could not be easily found using the DOAJ search.

To our knowledge, this is the first report showing that the availability of open access options depends on the source of funding. A previous study by Solomon and Björk analysing the source of funding for open access publishing across 74 open access journals of different disciplines showed that 50% of the open access publications in health sciences, biology and life sciences were

**Table 3** Open access policies of journals with high impact factors that offer immediate open access with the CC BY licence (n=21)

| Publisher | Organisation status | Journals included (n=21) | Open access variants available* | | | Funding requirements for obtaining open access with a CC BY licence |
| --- | --- | --- | --- | --- | --- | --- |
| | | | Embargo period† | Creative commons licence | Version of article available | |
| American Association for the Advancement of Science | Non-profit society | *Science; Sci Transl Med* | None | CC BY | VoR | The american association for the advancement of science 'will allow authors funded by the Bill & Melinda Gates Foundation to publish their research with a CC BY licence'‡ |
| | | | None | None | Accepted | |
| | | | 6 months | None | Accepted | |
| | | | 12 months | None | VoR | |
| American Society of Clinical Oncology | Non-profit society | *J Clin Oncol* | None | CC BY CC BY-NC-ND | VoR | Creative commons licences available only if funders are 'academical institutions, not-for-profit organisations, philanthropical foundations or government agencies' |
| | | | 6 months | None | VoR | |
| | | | 12 months | None | VoR | |
| BMJ Publishing Group | Non-profit society | *BMJ* | None | CC BY CC BY-NC | VoR | CC BY licence available for authors 'where the funder requires it' |
| Cell Press | Commercial | *Cancer Cell; Cell; Cell Metab; Cell Stem Cell; Immunity* | None | CC BY CC BY-NC-ND | VoR | Creative commons licences 'available only to authors covered by a funding body agreement' (these non-commercial funding bodies are listed on the journal websites) |
| | | | 12 months | None | Accepted | |

Continued

**Table 3** Continued

| Publisher | Organisation status | Journals included (n=21) | Open access variants available* | | | Funding requirements for obtaining open access with a CC BY licence |
|---|---|---|---|---|---|---|
| | | | Embargo period† | Creative commons licence | Version of article available | |
| Elsevier | Commercial | Eur Urol; Gastroenterology; J Am Coll Cardiol; Lancet; Lancet Diabetes Endocrinol; Lancet Infect Dis; Lancet Oncol; Lancet Neurol; Lancet Respir Med | None; 6 months | CC BY; CC BY-NC-ND; None | VoR; Accepted§/VoR; VoR | Creative commons licences are available to authors funded by specific funding bodies (these non-commercial funding bodies are listed on the journal websites) Elsevier has established agreements and developed policies to allow authors who publish in Elsevier journals to comply with manuscript archiving requirements of various funding bodies (these non-commercial funding bodies are listed on the journal websites) |
| European Society of Cardiology | Non-profit society | Eur Heart J | None; None; 12 months | CC BY; CC BY-NC; CC BY-NC-ND; None; None | VoR; Accepted; Accepted | 'RCUK/Wellcome Trust-funded authors...can use the CC BY licence for their articles' |
| Lippincott Williams & Wilkins | Commercial | Circulation | None; 6–12 months | CC BY; CC BY-NC; CC BY-NC-ND; None | VoR; Accepted | "Note that authors funded by RCUK or the Wellcome Trust may choose the CC BY licence if they agree to pay the article processing charge and commercial reuse of the article is not a factor' |

**Table 3** Continued

| Publisher | Organisation status | Journals included (n=21) | Open access variants available* | | | Funding requirements for obtaining open access with a CC BY licence |
|---|---|---|---|---|---|---|
| | | | Embargo period† | Creative commons licence | Version of article available | |
| Wiley-Blackwell | Commercial | CA Cancer J Clin | None | CC BY CC BY-NC CC BY-NC-ND | VoR | 'All RCUK and Wellcome Trust-funded authors will be directed to the CC BY licence' |
| | | | 12–24 months | None | Accepted | |

CC BY, Creative Commons Attribution; NC, Non-Commercial; ND, No Derivatives; RCUK, Research Councils UK; VoR, version of record.
*Available under the terms specified on the journal website.
†None = immediate open access; >0 months = delayed open access.
‡The American Association for the Advancement of Science's pilot open access partnership with the Gates Foundation concluded on 30 June 2018.[39]
§Accepted manuscripts can be self-archived and are required to attach a CC BY-NC-ND licence.[25]

funded by a grant/contract or national funding and 30% of the publications were funded by an institution.[41] However, the study did not show that the availability of open access was dependent on whether the source of funding is commercial or non-commercial.[41] In line with our results, the analysis by Solomon and Björk showed that journals with the highest impact factor tended to charge the highest article processing charges.[41] Limitations on the availability of the CC BY licence depending on the research funder are not in line with statements such as the Budapest Declaration,[17] the Berlin Declaration[18] and the Bethesda Statement,[19] which aim to provide end users with immediate access to research articles and to give them the opportunity to reuse material without restrictions. Furthermore, placing restrictions on access to medical research owing to its source of funding is not in line with the key principles of human research ethics laid out in the Declaration of Helsinki.[42]

Good Publication Practice 3 (GPP3) guidelines state that authors should take responsibility for the way research findings are published.[43] In line with these recommendations, commercial companies can and, we believe, should advise authors to reach a consensus on which journal to publish with, to avoid predatory journals and to adhere to sponsor guidelines and regulations. In the authors' experience, some pharmaceutical companies already have internal guidelines recommending open access publishing, and two (Shire, now part of Takeda, and Ipsen) now requires it.[37 38]

Our research shows that one-third of the journals with a high impact factor do not offer immediate access to the published version of a manuscript on publication, even though the open access policies of many funders with respect to embargo periods echo the recommendations set out by open access declarations worldwide.[17–19 27–29 32 44] Of note, Horizon 2020, which is supported by the European Research Council, requires its beneficiaries to make publications open access no later than 6 months after the official publication date and to make every effort to allow for maximum reuse of the materials, whether that be copying, distributing, searching, linking, crawling, mining or some other use.[45 46] Furthermore, cOAlition S, a group of national research funders with the support of the European Commission and the European Council, has committed to Plan S, the key principle of which is that scientific publications on research funded by participating national and European funders must be published open access by 2021.[44] Under the terms of Plan S, authors must retain copyright of their publication with no restrictions, and all publications must be published under an immediate open licence (preferably CC BY) that fulfils the requirements defined by the Berlin Declaration.[44 45]

Policies vary between publishers but also across journals at the same publisher, and this is also the case for journals not included in this analysis, as shown, for example, by Taylor & Francis in their table of the policies of all their journals.[47] Differences in policy have many underlying factors, including the choices of the journals' academical

editorial boards and societies. A potential disincentive to publishers offering CC BY licences to commercial research funders is the revenue generated from copyright fees and reprints. Permission to reproduce copyrighted materials can cost hundreds or even thousands of dollars; for example, the permission fee requested for reuse of a single table containing 40 words in the journal *American Family Physician* was US$4400.[48] Reprints can cost significantly more than permissions charges; for example, reprint sales from a single clinical trial can total US$1 million or more, with a large profit margin.[49]

Research by Lundh *et al*[50] aimed to quantify reprint revenues as a proportion of journal income. Of the six journals investigated, the two European journals, *The BMJ* and *The Lancet*, owned by Elsevier, disclosed the information requested. The editors of the US journals *Archives of Internal Medicine*, *Annals of Internal Medicine*, *JAMA* and the *New England Journal of Medicine* did not provide the data. For *The BMJ*, reprint revenues constituted 3% of its overall income; *The Lancet* obtained 41% of its revenue from reprints.[50] In *The Lancet*, commercially funded publications constituted a large proportion of highly reprinted articles (63/88) compared with a sample of control articles from the same journal (23/88).[51] The generation of revenue for publishers from the selling of reprints leaves publishers open to the criticism that bias can be introduced into editorial decisions.[50] This concern could be addressed by a transition to open access publishing exclusively with a CC BY licence.

Two of the journals included in our analysis, *Science* and *Science Translational Medicine*, both published by the American Association for the Advancement of Science, do not disclose article processing charges on their websites[39]; instead, they provide this information on their acceptance of an article. This practice does not comply with the DOAJ guidelines,[52] which state that processing fees must be stated clearly on journal websites in a place that is easy to find for potential authors prior to submitting their manuscript.

We found that the open access policies of some journals precluded commercially funded research from being published open access, even after an embargo period and without a Creative Commons licence. Further analyses could therefore be undertaken to clarify the proportion of journals with this policy and the rationale behind this position. Future research could also focus on a larger cohort of journals than the current study, or on journals from a specific therapy area, to clarify further the use of open access variants in the medical publications landscape. Future work could also involve collecting information on whether medical journals with a high impact factor allow commercial funders to use preprints or registered reports, which speed up research dissemination and remove publication bias, respectively. For example, it would be interesting to see whether journals that do not provide immediate open access options to commercial funders allow research manuscripts to be posted as preprints, and therefore support immediate dissemination of the results, although in a manuscript that has not yet been peer reviewed.

## CONCLUSIONS

The CC BY licence is recommended by open access declarations and funders of research as the optimal open access licence. Our analysis shows that although medical journals with a high impact factor provide some form of open access, they restrict commercially funded research from being published with the CC BY licence, meaning that the research output cannot be reused or built on if it is published in journals with a high impact factor without payment of additional fees. These restrictions hamper the further development and implementation of the approximately half of all medical research that is funded by commercial research funders.[1 33 34]

Open access publishing facilitates faster and more thorough disclosure of research, removes barriers for groups conducting systematic reviews, increases both the citation counts and altmetric scores of publications and benefits patient health by improving informed decision-making by doctors and patients.[9] Commercial research funders lag behind non-commercial funders in the implementation of open access policies, and we believe that it is time for them to close the gap. Commercial companies could, and we believe should, make clear their open access requirements, for example in a unified position statement, ideally aligned with open access declarations,[17–19] the Horizon 2020 programme and Plan S,[44–46] and the International Committee of Medical Journal Editors[53] and GPP3[43] guidelines, and then work together with publishers to realise the ultimate goal of improved access to medical research for all.

**Acknowledgements** Robert Kiley (https://orcid.org/0000-0003-4733-2558) is Head of Open Research at the Wellcome Trust, London, UK, and contributed to the review of this manuscript. Paul Farrow (https://orcid.org/0000-0002-0569-9688) is an employee of Oxford PharmaGenesis, Oxford, UK, and contributed significantly to the review of this manuscript. Sarah Stokes (https://orcid.org/0000-0002-8761-8588) and Velissaria Vanna are employees of Oxford PharmaGenesis, Oxford, UK, and contributed to the review and editing of this manuscript. The authors also thank Alan Thomas and Elizabeth Kinder for their review of this article from the patient perspective. This work was presented as a poster at both the European Meeting of the International Society for Medical Publication Professionals (ISMPP) on 23 January 2018 and the Annual Meeting of ISMPP on 2 May 2018 and was posted to bioRxiv as a preprint on 22 January 2018 (https://www.biorxiv.org/content/early/2018/01/22/250613).

**Contributors** Conceptualisation, project administration, TE (https://orcid.org/0000-0003-0307-725X), TK (https://orcid.org/0000-0001-6152-7365), LS (https://orcid.org/0000-0001-6117-781X), AW (https://orcid.org/0000-0002-9354-6402); methodology, resources, investigation, formal analysis, TE; writing – original draft, TE and LS; visualisation, TE; writing – review and editing, TE, TK, LS, AW, CW (https://orcid.org/0000-0003-3267-3990); supervision, TK, LS.

**Funding** This research was funded by Oxford PharmaGenesis.

**Competing interests** Tim Ellison, Tim Koder and Christopher Winchester are employees of Oxford PharmaGenesis, Oxford, UK. At the time of the research and writing of this manuscript, Laura Schmidt and Amy Williams were employees of Oxford PharmaGenesis, Oxford, UK, and are currently employed by Comradis and dna Communications, respectively. Christopher Winchester is also a Director and a shareholder of Oxford PharmaGenesis Holdings Ltd.

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
