## [Reviewer comments · BMJ Open]

ARTICLE DETAILS

TITLE (PROVISIONAL)	Open access policies of leading medical journals: a cross-sectional study
AUTHORS	Ellison, Tim; Koder, Tim; Schmidt, Laura; Williams, Amy; Winchester, Chris

VERSION 1 - REVIEW

REVIEWER	Jon Tennant IGDORE, UK
REVIEW RETURNED	08-Jan-2019

GENERAL COMMENTS	Dear Editor, Herein is my response to manuscript ID bmjopen-2018-028655, entitled Open access policies of leading medical journals: a cross-sectional study, by Tim Ellison and colleagues to the journal BMJ Open. The authors present an analysis of OA policies in a select group of 'high impact' medical journals. In particular, they look at how frequent CC BY licensing is a part of OA policies at these journals. This research is extremely timely, given the vigorous debates happening in the OA world at the present. As such, I expect this article to be able to help inform debates by filling a knowledge gap, as well as potentially being relevant to changes in the OA policy landscape at a variety of levels. This research is well-suited for the remit of the journal. My relevant expertise in reviewing this manuscript comes from being a researcher deeply interested in developments in Open Access. As such, I am deeply interested in seeing that papers like this get published to encourage the further discussion on the matter. That being said, there are several small issues I have with the data/methods that I am sure the authors will be able to easily address. I also have several suggestions for making this research feel more complete, that might require some extra data gathering. However, I don't want to be 'that reviewer', and note that these are only suggestions, and while they would make this research perhaps more impactful and comprehensive, should not preclude the publication of this work. Basic reporting Figures • The figures are all relevant, legible, and well-integrated into the text.
---

Data

- The supporting data are included with the manuscript in the supplementary information file. My only comment here is that the tables might be of more use as a csv file, rather than embedded within a docx file.
- I also have a slight concern with the publishing of email responses as part of the data collection for this study. Was permission sought in each case to publish the content of the emails? If not, this might need to be addressed post hoc prior to publication.

General comments

- This is a well-written article, and very timely given the current and ongoing debates around OA.
- One concern that popped up right away, given the context of this research, is that it was not looked at whether the journals in this small sample had policies encouraging/allowing the use of preprints (to speed up research dissemination), or registered reports (to remove publication bias), both of which seem highly relevant here.

Abstract

- The abstract is concise and conveys the context and main findings of the research.
- In the results subsection, it might need clarifying what an embargo period is for readers who might not be familiar with this aspect of OA.
- In the strengths/limitations, it needs to be made clear again that this is for medical journals only. For point 3, I would say might change in the future, as opposed to not being current. Also, is this study really systematic if it excludes a large number (the majority of?) journals (i.e., those with an IF less than 15). Are all the journals selected also exclusively English language only?
- I disagree with the final point here too, that extending this analysis would be cumbersome. It seems that the methods are based around simple Web searches (with supplementary emails), which while possibly tedious to perform, certainly would not be difficult to achieve. I have done something similar for more than 200 journals in Palaeontology, and while it did take some effort, was relatively simple to perform (https://paleorxiv.github.io/journal_policies.html). Maybe I should try and get this published too... 😊 So I would say it is possible, but would just take more time, as well as increasing the scope of the study. None of this precludes the inherent value of this research though.

Introduction

- Note that some of the research funders are also philanthropic (e.g., Bill and Melinda Gates Foundation).
- I would add that, for extra context, 'traditional publishing' stifles use of research by actively prohibiting re-use (e.g., through copyright), as well as through paywalls and the corrupting effect that it has on selective communication of research (e.g., only communication of 'positive' effects). Some of this is not resolved purely through OA.
- Need to be careful here to distinguish between altmetrics, as in alternative metrics, and Altmetric the company, which provides altmetrics as a service.
- Some might argue that OA has one true definition, more akin to CC BY or equivalent, and that variations of this amount to a

form of 'open washing'; for example, bronze OA or 'free to read' OA don't qualify as 'true OA' given some definitions.

- Might be worth adding here, just for completeness, what the licensing requirements for indexing in the DOAJ are (https://doaj.org/application/new#license_url-container).

- I disagree with the statement 'Open access with a Creative Commons licence is typically facilitated by article processing charges', for several reasons. Firstly, OA publication via 'green' self-archiving does not require APCs. Secondly, there are 1000s of journals within the DOAJ that do not leverage APCs and publish with CC licenses. I wouldn't therefore say it is typical, but is perhaps a common feature typical for certain publishers (especially the 'big five').

- Also, worth noting that accepted manuscripts or postprints are often required to have CC licenses attached, although often with NC and/or ND clauses attached. It's all a bit messy, so just be careful with the wording here. See, for example, Elsevier (<https://www.elsevier.com/about/policies/sharing>).

- The following paper seems highly relevant here: <https://health-policy-systems.biomedcentral.com/articles/10.1186/s12961-017-0235-3>

- It describes the uptake of OA in global health research, and includes statistics such as "60.8% of researchers do not self-archive their work even when it is free and in keeping with journal policy."

- Could it be made clearer exactly what is meant by a 'commercial' funder here?

Materials and methods

- The methods are clearly explained and simple enough to follow.

- I feel another data column could be added here, on whether the journals are owned/published by a commercial or non-profit publisher.

Results

- The results are presented in a clear and logical fashion. Is there any reason given for the threshold used for the impact factor groupings?

- Here, as I mentioned above too, it is worth checking whether the 'embargoed OA' policies include reference to a CC license. And also maybe making the distinction clearer between immediate and delayed OA.

- Not going to say that I didn't frown immensely at the APCs quoted here. Is there any justification given to the APCs in any of the journals? Furthermore, is there any correlation between APC and IF in this sample?

Discussion

- Critically, in the first paragraph here, it also means that the vast majority of journals would be non-compliant with commercial funders who had CC BY requirements in their OA policies. So, you get a sort of chicken or the egg scenario. Do the funders impose these requirements, and potentially compromise publication venue choice, or do journals change their policies first?

- So, in terms of a 'larger scale' analysis, it is possible to do a quick and dirty comparison using the DOAJ. For example, here is a search of 700+ medical journals in the DOAJ, with relevant license information, and whether or not they charge APCs. I think it

would be eminently feasible to include these data as a comparison within the present study.

- Sadly, I also do not think this study is the first to look at OA and funding sources. See, for example, Solomon and Bjork (2012): <https://onlinelibrary.wiley.com/doi/abs/10.1002/asi.21660> (and references therein, as well as some citing articles). Indeed, the discussion section does not seem to explore much of the literature on policies of OA journals to place this study into a wider context.

- That anecdote about the American Family Physician is incredible. I think it is worth highlighting this a bit more, that having commercial entities governing/owning rights to research is directly anathema to the access and re-use of that research for societal goods. And indeed that this remains one of the biggest tensions within the current scholarly publishing 'marketplace'. Especially as one of the key aspects of the open access movement has been about democratizing access to knowledge, which is seen as a major threat to commercial publishers (e.g., <https://www.norrag.org/democratising-knowledge-a-report-on-the-scholarly-publisher-elsevier-by-dr-jonathan-tenant/>).

- Worth noting that the Lancet family of journals are now owned by Elsevier?

- "This concern could be addressed by a transition to open access publishing exclusively with a CC BY licence. However, such a transition may need to be managed." Could this be expanded on, as it seems a bit vague to me.

- Again, the anecdote with the AAAS is incredible. Is it worth highlighting that they are supposed to be the most prestigious learned society, and that this regressive attitude/practice around OA seems very counter-intuitive? This is also not the first time in which the AAAS have been called out for things like this... I would, however, be very careful on the association of this behavior with predatory journals. I don't think the AAAS would be too pleased at being alluded to as predatory, especially for their flagship journal. Perhaps this just needs to be more carefully worded.

- Also, isn't Science an interdisciplinary journal, rather than a medical journal? I wonder if this could be a potential confounding factor for some of the results.

Conclusions

- Just a thought here; might it be that commercially-funded research is seen to potentially have the most commercial value (as opposed to e.g., blue skies research), and thus there is a bigger incentive for commercial publishers to 'capture' this IP, restrict access or open licensing for it, so that they can maximise revenues resulting from it?

- "However, there are concerns that a rapid transition to publishing exclusively with a CC BY licence will be difficult, given current processes and business models in scientific publishing." Could you possibly mention a couple of these concerns?

- For the final paragraph, I'm not sure the idea that OA is beneficial to some publishers (as one of the stakeholders) is that strong. Which is perhaps why some of the strongest resistance to OA has historically come from commercial publishers, as well as societies who are largely dependent on revenue from subscriptions. Indeed, the discord you are seeing between funding policies, what researchers want/need, and what is good for science and what the scholarly publishing industry provides is probably at least in part a reflection of this.

- The last statement here comes across as an advocacy point. I would tone this down to make it more of a suggestion, and

	perhaps include a list of several clear points that could be taken to achieve this. Congratulations to the authors on a great piece of work, and I look forward to seeing their research published. Sincerely, Jonathan Tennant
--	--

REVIEWER	Carly Strasser Fred Hutchinson Cancer Research Center, USA
REVIEW RETURNED	27-Jan-2019

GENERAL COMMENTS	The paper is well written and reports on important findings. My one request is that the authors find ways to emphasize the commercial funder point more. That is, sometimes it's called "industry funding", sometimes "commercial", and sometimes "pharmaceutical". I would suggest picking one phrase, defining in the intro, and sticking to it throughout the manuscript for clarity. I am only somewhat familiar with how medical research is funded, and was surprised to learn that 50% comes from industry. This is an important component of the story! I would also suggest moving up the reprint profits as a mention in the introduction. This part of the discussion made me want to go back and re-read (which is a testament to the importance of the results!). One other comment is the use of "Altmetric" as shorthand for article-level metrics. Although the company Altmetric does have handy ways to report scores, the article-level metrics scene is much bigger than them (e.g., PLOS was the inventor; see https://www.plos.org/article-level-metrics). Two very minor edits/suggestions: Page 9 Line 46: double "only" Page 9 Line 53: the statement that "other journals might be inclined to change their policy" creeps into a suggestion/criticism for journals; might consider rewording to maintain impartiality.
---

VERSION 1 – AUTHOR RESPONSE

Reviewers' comments

Reviewer 1: Carly Strasser

The paper is well written and reports on important findings. My one request is that the authors find ways to emphasize the commercial funder point more. That is, sometimes it's called "industry funding", sometimes "commercial", and sometimes "pharmaceutical". I would suggest picking one phrase, defining in the intro, and sticking to it throughout the manuscript for clarity.

Response

We thank Dr Strasser for her positive response to our manuscript and findings. As suggested, we have standardized the references to 'commercial' funding throughout the manuscript. Now, all mentions of 'commercial' funding have been described as such and we have clarified in the

penultimate paragraph of the Introduction that 'commercial' research funders are defined as pharmaceutical companies and other medical industries that fund research for commercial purposes.

I am only somewhat familiar with how medical research is funded, and was surprised to learn that 50% comes from industry. This is an important component of the story! I would also suggest moving up the reprint profits as a mention in the introduction. This part of the discussion made me want to go back and re-read (which is a testament to the importance of the results!).

Response

Although we agree with Dr Strasser that the reprint revenue of journals is important for understanding our findings, we do not think the Introduction section is a suitable place to mention them up front. Reprint revenue was considered as a possible explanation for our results in the Discussion section and we do not want to mislead readers into thinking we pre-empted its importance.

One other comment is the use of "Altmetric" as shorthand for article-level metrics. Although the company Altmetric does have handy ways to report scores, the article-level metrics scene is much bigger than them (e.g., PLOS was the inventor; see <https://www.plos.org/article-level-metrics>).

Response

We thank Dr Strasser for her comment and have changed 'Altmetric scores' to 'altmetric scores' to use the more generalized term of altmetrics rather than the company 'Altmetric'.

Two very minor edits/suggestions:

Page 9 Line 46: double "only"

Page 9 Line 53: the statement that "other journals might be inclined to change their policy" creeps into a suggestion/criticism for journals; might consider rewording to maintain impartiality.

Response

We have deleted the first 'only' in the sentence referenced by Dr Strasser.

We have reworded the sentence "The BMJ would be suitable, and other journals might be inclined to change their policy" to:

"...The BMJ would currently be the only compliant medical journal with an impact factor greater than 15.0"

Reviewer 2: Jon Tennant

Dear Editor,

Herein is my response to manuscript ID bmjopen-2018-028655, entitled Open access policies of leading medical journals: a cross-sectional study, by Tim Ellison and colleagues to the journal BMJ Open.

The authors present an analysis of OA policies in a select group of 'high impact' medical journals. In particular, they look at how frequent CC BY licensing is a part of OA policies at these journals. This research is extremely timely, given the vigorous debates happening in the OA world at the present. As such, I expect this article to be able to help inform debates by filling a knowledge gap, as well as potentially being relevant to changes in the OA policy landscape at a variety of levels. This research is well-suited for the remit of the journal.

My relevant expertise in reviewing this manuscript comes from being a researcher deeply interested in developments in Open Access. As such, I am deeply interested in seeing that papers like this get published to encourage the further discussion on the matter. That being said, there are several small issues I have with the data/methods that I am sure the authors will be able to easily address. I also have several suggestions for making this research feel more complete, that might require some extra data gathering. However, I don't want to be 'that reviewer', and note that these are only suggestions, and while they would make this research perhaps more impactful and comprehensive, should not preclude the publication of this work.

Response

We thank Dr Tennant for his comprehensive review of our manuscript and for his valuable feedback.

Basic reporting

Figures

- The figures are all relevant, legible, and well-integrated into the text.

Data

- The supporting data are included with the manuscript in the supplementary information file. My only comment here is that the tables might be of more use as a csv file, rather than embedded within a docx file.

- I also have a slight concern with the publishing of email responses as part of the data collection for this study. Was permission sought in each case to publish the content of the emails? If not, this might need to be addressed post hoc prior to publication.

Response

We did not receive permission from the journals to include the emails confirming our findings in our manuscript. We have therefore removed Table S2 from the Supplemental information and any reference to Table S2 in the manuscript text.

We have now uploaded the Supplemental information, which now only contains a table describing the Creative Commons licences, as a PDF file.

General comments

- This is a well-written article, and very timely given the current and ongoing debates around OA.
- One concern that popped up right away, given the context of this research, is that it was not looked at whether the journals in this small sample had policies encouraging/allowing the use of preprints (to speed up research dissemination), or registered reports (to remove publication bias), both of which seem highly relevant here.

Response

We thank Dr Tennant for his positive feedback.

We agree that it would have been insightful to additionally look at whether the analysed journals encouraged/allowed the use of preprints or registered reports. However, the objective of our study was to specifically analyse the open access policies of medical journals with a high impact factor and, in particular, the availability of the CC BY licence. We have added to the final paragraph of the Discussion section that future work could involve collecting information on whether medical journals with a high impact factor allow the use of preprints or registered reports, which speed up research dissemination and remove publication bias, respectively. We added as an example that it would be interesting to see whether journals that do not provide immediate open access options to commercial funders allow research manuscripts to be posted as preprints, and therefore support immediate dissemination of the results, although in a manuscript that has not yet been peer reviewed.

Abstract

- The abstract is concise and conveys the context and main findings of the research.
- In the results subsection, it might need clarifying what an embargo period is for readers who might not be familiar with this aspect of OA.

Response

We have removed mention of “embargo period” in the abstract to avoid confusion and clarified that all 35 journals offered some form of open access allowing articles to be free-to-read either immediately on publication or after a delay of up to 12 months.

- In the strengths/limitations, it needs to be made clear again that this is for medical journals only. For point 3, I would say might change in the future, as opposed to not being current. Also, is this study really systematic if it excludes a large number (the majority of?) journals (i.e., those with an IF less than 15). Are all the journals selected also exclusively English language only?

Response

We have updated the strengths/limitations and Discussion sections to accommodate Dr Tennant’s suggestions. We have changed “systematic” to “cross-sectional” throughout the manuscript.

The journals were selected using Sylogent’s Journal Selector. The journals could have been in a language other than English, but all journals that met the search criteria were in English.

- I disagree with the final point here too, that extending this analysis would be cumbersome. It seems that the methods are based around simple Web searches (with supplementary emails), which while possibly tedious to perform, certainly would not be difficult to achieve. I have done something similar for more than 200 journals in Palaeontology, and while it did take some effort, was relatively simple to perform (https://paleorxiv.github.io/journal_policies.html). Maybe I should try and get this published too... 😊 So I would say it is possible, but would just take more time, as well as increasing the scope of the study. None of this precludes the inherent value of this research though.

Response

We agree with Dr Tennant’s suggestion and have updated the limitation in question in the strengths/limitations and Discussion sections to say that an extended manual analysis would take more time and increase the scope of the study.

Introduction

- Note that some of the research funders are also philanthropic (e.g., Bill and Melinda Gates Foundation).

Response

We have added ‘philanthropic organizations’ to the first sentence.

- I would add that, for extra context, 'traditional publishing' stifles use of research by actively prohibiting re-use (e.g., through copyright), as well as through paywalls and the corrupting effect that it has on selective communication of research (e.g., only communication of 'positive' effects). Some of this is not resolved purely through OA.

Response

We have added to the first paragraph of the Introduction: "Implementation of research published using the traditional subscription publication model is hindered by copyright restrictions that prohibit reuse of the published content and paywalls that prevent public access." We have not commented on selective communication of research because this would not be affected by open access publishing.

- Need to be careful here to distinguish between altmetrics, as in alternative metrics, and Altmetric the company, which provides altmetrics as a service.

Response

We agree with Dr Tennant's comment and have changed 'Altmetric scores' to 'altmetric scores' to use the more generalized term of altmetrics rather than the company 'Altmetric'.

- Some might argue that OA has one true definition, more akin to CC BY or equivalent, and that variations of this amount to a form of 'open washing'; for example, bronze OA or 'free to read' OA don't qualify as 'true OA' given some definitions.

Response

We have added to the third paragraph of the Introduction: "The Budapest Open Access Initiative, the Berlin Declaration, the Bethesda Statement and open access advocates define 'open access' exclusively as published content that can be read free-of-charge immediately at the time of publication with unrestricted reuse rights providing that the original source is attributed." To accommodate this change, we have reordered and reworded some of the Introduction text.

- Might be worth adding here, just for completeness, what the licensing requirements for indexing in the DOAJ are (https://doaj.org/application/new#license_url-container).

Response

We have added the licensing requirements for indexing in the DOAJ to the end of the fourth paragraph of the Introduction.

- I disagree with the statement 'Open access with a Creative Commons licence is typically facilitated by article processing charges', for several reasons. Firstly, OA publication via 'green' self-archiving does not require APCs. Secondly, there are 1000s of journals within the DOAJ that do not leverage APCs and publish with CC licenses. I wouldn't therefore say it is typical, but is perhaps a common feature typical for certain publishers (especially the 'big five').

Response

We thank Dr Tennant for his clarification on this point and have preceded the sentence with "At prominent journals...".

- Also, worth noting that accepted manuscripts or postprints are often required to have CC licenses attached, although often with NC and/or ND clauses attached. It's all a bit messy, so just be careful with the wording here. See, for example, Elsevier (<https://www.elsevier.com/about/policies/sharing>).

Response

We have added the following sentence to the end of the fifth paragraph of the Introduction. "It is noteworthy that the accepted version of a manuscript as well as the Version of Record are sometimes required to bear a Creative Commons licence, often including the -NC and/or -ND clause."

- The following paper seems highly relevant here: <https://health-policysystems.biomedcentral.com/articles/10.1186/s12961-017-0235-3>

o It describes the uptake of OA in global health research, and includes statistics such as "60.8% of researchers do not self-archive their work even when it is free and in keeping with journal policy."

Response

We agree with Dr Tennant that the Smith et al. paper is highly relevant and thank him for sharing it with us. We have added the following sentence to the penultimate paragraph of the Introduction.

"However, a study analysing global health research articles published between 2010 and 2014 showed that 69% of these articles were not freely available on the journal's website and 61% of researchers do not self-archive their work even when journal policy allows them to do so free of charge".

- Could it be made clearer exactly what is meant by a 'commercial' funder here?

Response

We have clarified in the penultimate paragraph of the Introduction that 'commercial' research funders are defined here as pharmaceutical companies and other medical industries that fund research for commercial purposes.

Materials and methods

- The methods are clearly explained and simple enough to follow.
- I feel another data column could be added here, on whether the journals are owned/published by a commercial or non-profit publisher.

Response

We thank Dr Tennant for his suggestion and have updated Table 2 and Table 3 with an extra column, 'Organisation status', showing whether the journals are published by a commercial or non-profit publisher.

Results

- The results are presented in a clear and logical fashion. Is there any reason given for the threshold used for the impact factor groupings?

Response

The impact factor groupings were chosen as such to have roughly similar numbers of journals in each category (IF 15.0–19.9 = 15 journals; IF 20.0–29.9 = 10 journals; IF ≥ 30 = 10 journals).

- Here, as I mentioned above too, it is worth checking whether the 'embargoed OA' policies include reference to a CC license. And also maybe making the distinction clearer between immediate and delayed OA.

Response

Of the journals analysed, the Elsevier-owned journals required the attachment of a CC BY-NC-ND licence to self-archived accepted manuscripts. We have added this information as a footnote to Table 3. No other journals analysed mentioned attaching a Creative Commons licence to articles published with delayed open access.

Regarding making the distinction clearer between immediate and delayed open access, we have added a footnote to the tables on the 'Embargo period' column: "None=immediate open access; > 0 months=delayed open access."

- Not going to say that I didn't frown immensely at the APCs quoted here. Is there any justification given to the APCs in any of the journals? Furthermore, is there any correlation between APC and IF in this sample?

Response

For all journals except Elsevier-owned journals, there was no obvious written justification for the article processing charges quoted on the journal websites. For Elsevier-owned journals, charges are said to be based on criteria including journal quality, editorial and technical processes, competitive considerations, market conditions and other revenue streams associated with the journal (<https://www.elsevier.com/about/policies/pricing>).

In our sample there was no obvious correlative trend between article processing charges and impact factors, with most of the journals that offered a CC BY licence and disclosed the article processing cost charging \$5000 and the sample being too small for statistically meaningful results. Of the journals that offered a CC BY licence and disclosed an article processing charge of less than \$5000, 5/6 had an impact factor of less than 20.0; one journal charging less than \$5000 had the highest impact factor of 137.6 and so was a prominent outlier in the dataset. We added a brief description of the 5/6 journals charging less than \$5000 to the Results section and compared our results to the Solomon and Bjork paper (see below).

Discussion

- Critically, in the first paragraph here, it also means that the vast majority of journals would be noncompliant with commercial funders who had CC BY requirements in their OA policies. So, you get a sort of chicken or the egg scenario. Do the funders impose these requirements, and potentially compromise publication venue choice, or do journals change their policies first?

Response

In response to another reviewer's concerns that we were not being impartial, we have toned down the sentence at the end of the first paragraph of the Discussion that said other journals may be inclined to change their policy if commercial companies mandate open access under a CC BY licence. In the Conclusion section of the manuscript, we suggest that commercial companies should make clear their open access requirements and work together with publishers to improve access to medical research. We feel that this would be the most suitable step forward and maintains our impartiality on the matter.

- So, in terms of a 'larger scale' analysis, it is possible to do a quick and dirty comparison using the DOAJ. For example, here is a search of 700+ medical journals in the DOAJ, with relevant license information, and whether or not they charge APCs. I think it would be eminently feasible to include these data as a comparison within the present study.

Response

We thank Dr Tennant for his advice and have carried out the DOAJ search suggested by him. We included the methods and results in their corresponding sections and compared the results with those from our manual analysis in the second paragraph of the Discussion.

- Sadly, I also do not think this study is the first to look at OA and funding sources. See, for example, Solomon and Bjork (2012): <https://onlinelibrary.wiley.com/doi/abs/10.1002/asi.21660> (and references therein, as well as some citing articles). Indeed, the discussion section does not seem to explore much of the literature on policies of OA journals to place this study into a wider context.

Response

We still believe that our study is the first to show that the availability of open access options depends on the source of funding. To make this point stronger, we have described the results of the Solomon and Bjork paper relating to funding source in the Discussion and clarified that their paper did not show the availability of open access depends on whether the source of funding is commercial or non-commercial. We carried out literature searches for other papers looking at funding source and open access and did not find any other suitable papers to reference. To help place our results in wider context, we added that Solomon and Bjork showed that journals with the highest impact factor tended to charge the highest article processing charges.

- That anecdote about the American Family Physician is incredible. I think it is worth highlighting this a bit more, that having commercial entities governing/owning rights to research is directly anathema to the access and re-use of that research for societal goods. And indeed that this remains one of the biggest tensions within the current scholarly publishing 'marketplace'. Especially as one of the key aspects of the open access movement has been about democratizing access to knowledge, which is seen as a major threat to commercial publishers (e.g., <https://www.norrag.org/democratising-knowledge-a-report-on-the-scholarly-publisher-elsevier-by-dr-jonathan-tennant/>).

Response

We thank Dr Tennant for his suggestion. However, we would prefer not to introduce too many of our own opinions into this manuscript and to leave our manuscript discussion focussed on the misalignment of journals' policies with open access declarations and guidelines, and on the discrepancy between the open access options offered to commercial and those offered to non-commercial funders.

- Worth noting that the Lancet family of journals are now owned by Elsevier?

Response

We have added "owned by Elsevier" after the first mention of The Lancet in the Discussion paragraph about reprint revenues.

- “This concern could be addressed by a transition to open access publishing exclusively with a CC BY licence. However, such a transition may need to be managed.” Could this be expanded on, as it seems a bit vague to me.

Response

We agree that this sentence is vague and have therefore deleted “However, such a transition may need to be managed”. We feel that it is not needed.

- Again, the anecdote with the AAAS is incredible. Is it worth highlighting that they are supposed to be the most prestigious learned society, and that this regressive attitude/practice around OA seems very counter-intuitive? This is also not the first time in which the AAAS have been called out for things like this... I would, however, be very careful on the association of this behavior with predatory journals. I don't think the AAAS would be too pleased at being alluded to as predatory, especially for their flagship journal. Perhaps this just needs to be more carefully worded.

Response

We have deleted the sentence suggesting that the AAAS journals are engaging in a practice common amongst predatory journals. We would prefer not to add any further opinions of our own on this matter to remain as impartial as possible.

- Also, isn't Science an interdisciplinary journal, rather than a medical journal? I wonder if this could be a potential confounding factor for some of the results.

Response

We agree that Science is an interdisciplinary journal rather than exclusively a medical journal. We have added that this is a potential confounding factor for some of the results in the Strengths/limitations and Discussion sections.

Conclusions

- Just a thought here; might it be that commercially-funded research is seen to potentially have the most commercial value (as opposed to e.g., blue skies research), and thus there is a bigger incentive for commercial publishers to 'capture' this IP, restrict access or open licensing for it, so that they can maximise revenues resulting from it?

Response

We agree with Dr Tennant that there is a bigger incentive for commercial publishers to profit from commercially-funded research because it potentially has the most commercial value. We feel that we have alluded to this point when discussing reprint revenues of journals.

- “However, there are concerns that a rapid transition to publishing exclusively with a CC BY licence will be difficult, given current processes and business models in scientific publishing.” Could you possibly mention a couple of these concerns?

Response

We have decided to delete this sentence from the manuscript because we feel it is not needed.

- For the final paragraph, I’m not sure the idea that OA is beneficial to some publishers (as one of the stakeholders) is that strong. Which is perhaps why some of the strongest resistance to OA has historically come from commercial publishers, as well as societies who are largely dependent on revenue from subscriptions. Indeed, the discord you are seeing between funding policies, what researchers want/need, and what is good for science and what the scholarly publishing industry provides is probably at least in part a reflection of this.

Response

We agree with Dr Tennant and have deleted the sentence in question

- The last statement here comes across as an advocacy point. I would tone this down to make it more of a suggestion, and perhaps include a list of several clear points that could be taken to achieve this.

Response

We have rephrased these statements so that they read more as suggestions. We believe that a unified position statement that involves publishers is the first step that commercial companies could take and therefore would prefer to leave this as the main action that could be taken to improve access to medical research.

Congratulations to the authors on a great piece of work, and I look forward to seeing their research published.

Response

We thank Dr Tennant for his congratulations and for his valuable review that has helped us improve our manuscript.

Sincerely,

Jonathan Tennant